# Update on the Symptomatic Treatment of Huntington’s Disease: From Pathophysiology to Clinical Practice

**DOI:** 10.3390/ijms26136220

**Published:** 2025-06-27

**Authors:** Gonzalo Olmedo-Saura, Eugenio Bernardi, Lidia Bojtos, Saül Martínez-Horta, Javier Pagonabarraga, Jaime Kulisevsky, Jesús Pérez-Pérez

**Affiliations:** 1Medicine Department, Universitat Autònoma de Barcelona (UAB), 08193 Barcelona, Spain; 2Movement Disorders Unit, Neurology Department, Hospital de la Santa Creu i Sant Pau, 08041 Barcelona, Spain; 3Institut d’Investigacions Biomèdiques-Sant Pau (IIB-Sant Pau), 08041 Barcelona, Spain; 4Centro de Investigación Biomédica en Red-Enfermedades Neurodegenerativas (CIBERNED), 28029 Madrid, Spain; 5European Huntington’s Disease Network (EHDN), 89081 Ulm, Germany

**Keywords:** Huntington disease, chorea, movement disorders, cognitive dysfunction, psychiatric symptoms, pathophysiology

## Abstract

Huntington’s disease (HD) is the most common autosomal dominant neurodegenerative disorder, characterized by a triad of motor dysfunction, cognitive decline, and psychiatric disturbances. While recent efforts have focused on developing disease-modifying therapies, no treatment has yet demonstrated clinical efficacy. As a result, symptomatic treatment remains the cornerstone of care. However, high-quality evidence from large randomized trials is limited, and therapeutic decisions must rely on clinical expertise and extrapolation from other neurological or psychiatric conditions. This narrative review provides a comprehensive and practical overview of symptomatic treatment strategies for HD with emphasis on the pathophysiological underpinnings of each symptom and the molecular mechanisms of available and emerging therapies, aiming to support rational, individualized management. Finally, we highlight the critical role of non-pharmacological interventions and the need for multidisciplinary approaches to optimize patient outcomes and quality of life.

## 1. Introduction

Huntington’s disease (HD) is the most common autosomal dominant neurodegenerative disorder, with an estimated prevalence of 10–12 cases per 100,000 individuals of European ancestry [1,2,3]. It is caused by an expansion of ≥36 CAG repeats (>39 for full penetrance) in the *HTT* gene [4], which encodes the huntingtin protein (HTT) [5,6]. The typical age at onset ranges from 30 to 60 years, with earlier onset and greater disease severity associated with longer expansions [7]. In addition to its well-characterized genetic basis, the clinical diagnosis and staging of HD have evolved from traditional motor-based criteria to a more biologically grounded framework. Classically, diagnosis is established in individuals with a pathogenic *HTT* expansion or positive family history who develop unequivocal motor signs, corresponding to a score of 4 (≥99% confidence) on the Diagnostic Confidence Level (DCL) of the Unified Huntington’s Disease Rating Scale (UHDRS) [5]. More recently, the Huntington’s Disease Integrated Staging System (HD-ISS) has been proposed for individuals with ≥40 CAG repeats. It defines four stages: stage 0 (gene expansion carriers without detectable disease-related changes), stage 1 (evidence of neurodegeneration, typically caudate and/or putaminal atrophy), stage 2 (emergence of motor and/or cognitive signs based on UHDRS-Total Motor Score (TMS) and Symbol Digit Modalities Test thresholds), and stage 3 (functional decline).

Although the pathophysiology of HD is not fully elucidated, it is established that the expanded CAG repeat results in the production of abnormal and unstable HTT species, including full-length mutant HTT (mHTT), exon 1 fragments, and products of repeat-associated non-ATG (RAN) translation [8,9]. These species promote neurodegeneration through toxic gain-of-function mechanisms, including protein misfolding, aggregation, RNA-mediated toxicity, and dysregulation of other cellular processes [10,11,12]. A key modulator of disease onset and progression is somatic instability of the CAG repeat expansion [13,14,15], particularly within vulnerable cell populations as the striatal medium spiny neurons (MSNs). This process is modulated by several DNA repair pathways, with strong evidence implicating mismatch repair genes as PMS1, MLH1, MSH3, PMS2, FAN1, LIG1 [6,14,15,16].

Clinically, HD is characterized by a triad of motor, cognitive, and psychiatric manifestations that reflect the progressive disruption of cortico-striato-thalamo-cortical circuits including the motor, associative, and limbic loops [17].

While numerous disease-modifying therapies are under development—with several already being evaluated in ongoing phase 1 and 2 clinical trials, and some projected to soon enter phase 3 [10,18]—the treatment of HD remains entirely symptomatic. The general guiding principle is to prioritize the most disruptive symptoms reported by patients and caregivers, while carefully accounting for comorbidities and the side effects (especially considering the heightened sensitivity of HD patients to medication side effects). Notably, most pharmacological interventions have not been evaluated in large-scale randomized clinical trials (RCTs), limiting the robustness of the evidence supporting their use. Consequently, an optimal therapeutic approach requires a careful balance between clinical expertise and evidence-based medicine [19].

The aim of this review is to provide an updated, comprehensive and practical overview of symptomatic treatment strategies for HD. Emphasis is placed on the pathophysiological rationale, mechanisms of action, and real-world applicability of pharmacological and non-pharmacological interventions across the motor, cognitive, and psychiatric domains of the disease. Disease-modifying therapies are beyond the scope of this review.

## 2. Search Strategy

For this narrative review, we conducted a non-systematic literature search of peer-reviewed articles published in English up to March 2025. The search was performed using the PubMed database. The following search strategy was used: (“Huntington Disease” [Title] OR “Huntington’s Disease” [Title] OR “Huntington*” [Title]) AND (“treatment” [Title/Abstract] OR “therapy” [Title/Abstract] OR “management” [Title/Abstract] OR “pharmacological” [Title/Abstract] OR “non-pharmacological” [Title/Abstract]) AND Humans [Mesh]. Additional references were identified by manually screening the reference lists of included articles and relevant reviews. Studies considered pertinent to the scope of this review were selected for inclusion.

## 3. Symptomatic Treatment of Huntington’s Disease

### 3.1. Motor Symptoms

HD was first described by George Huntington in 1872 as “hereditary chorea” [20]. However, the motor phenotype is more complex and evolves with disease progression, encompassing dystonia, gait abnormalities, motor impersistence, myoclonus, oculomotor disturbances and in advanced stages bradykinesia and rigidity.

These motor manifestations reflect the progressive disruption of basal ganglia circuits:

In early stages, there is preferential degeneration of D2-expressing GABAergic MSNs of the indirect pathway. These neurons, which contain enkephalin and project to the external globus pallidus (GPe), normally inhibit thalamocortical drive. Their loss leads to disinhibition of thalamic output and increased cortical excitability, resulting in hyperkinetic features such as chorea [17,21,22,23].

As neurodegeneration progresses to involve D1-expressing MSNs of the direct pathway and those that contain substance P and project to the internal globus pallidus (GPi), parkinsonian symptoms including bradykinesia and rigidity become more prominent [17,21,22,23].

**Juvenile-onset HD or juvenile-HD** (defined as onset before 21 years) accounts for 4–10% of cases [24] and typically presents with parkinsonism and dystonia as the initial motor symptoms, rather than chorea. This distinct phenotype, with more rapid progression, requires adapted symptomatic approaches [25,26,27].

This section will review the most clinically relevant motor symptoms of adult-onset HD individually, highlighting their pathophysiological basis and therapeutic management. Brief comments on juvenile HD are included where appropriate.

#### 3.1.1. Chorea

Chorea is the hallmark motor feature of HD and often the earliest observed manifestation. It is also the symptom for which the greatest number of targeted interventions have been studied and are currently available. It refers to involuntary, irregular, purposeless, nonrhythmic, abrupt, rapid, unsustained movements that seem to flow from one part of the body to another [28,29]. The key for differentiating from other hyperkinetic movements is its randomness in timing, direction, and distribution [28,30].

In early stages, chorea typically begins in the facial muscles, especially the forehead, later spreads to the distal extremities and progressively involves more proximal areas such as the trunk. Other hyperkinetic phenomena may co-occur, including athetosis (irregular writhing movements predominantly involving distal extremities), choreo-dystonia, and ballism (large, proximal choreatic movements) [29,31]. Nocturnal choreatic movements may represent one of the earliest motor manifestations [32,33]. Despite being visibly evident, patients frequently lack awareness of their movements, with anosognosia being common and clinically significant [34,35].

As previously described, chorea in HD results from underactivation of the indirect basal ganglia pathway due to early degeneration of D2-expressing MSNs [36,37]. Although simplified, a useful framework for understanding treatment strategies is that activation of D2 receptors (e.g., by endogenous dopamine) exacerbates chorea by suppressing the indirect pathway, whereas D2 antagonism can restore inhibitory tone and reduce hyperkinetic symptoms.

##### Clinical Management

Chorea does not always require treatment. In mild or non-disabling cases, supportive interventions such as patient and caregiver education, environmental modifications and adaptive tools may be enough. Pharmacologic therapy is generally reserved for cases causing functional impairment, loss of balance, discomfort, or leads to social stigma [30,38,39]. The goal of treatment is to reduce the severity of chorea—not to eliminate it completely—while minimizing side effects and improving quality of life.

Several therapeutic options are available; however, monotherapy should be prioritized, with combination therapy reserved for refractory cases, given the lack of robust evidence [38] and the increased risk of additive adverse events (e.g., QTc prolongation, neuroleptic malignant syndrome, sedation, cognitive worsening). A therapeutic decision-making algorithm for the treatment of chorea in HD is presented in Figure 1.

A.Vesicular Monoamine Transporter type 2 (VMAT2) inhibitors

The first pharmacological approach for chorea was reserpine [40], which reduced involuntary movements through irreversible VMAT inhibition but caused significant peripheral (VMAT type 1) and central (VMAT type 2) side effects due to non-selective blockade [41]. VMATs load monoamines—dopamine, serotonin, noradrenaline, and histamine—into synaptic vesicles and its inhibition reduces vesicular storage, diminished synaptic release, and increased cytosolic degradation.

This led to the development of tetrabenazine (TBZ), a reversible and selective VMAT2 inhibitor, which preferentially reduces dopaminergic signaling [42,43] and became the first drug specifically approved for HD chorea following its demonstrated efficacy in the TETRA-HD pivotal trial [44]. Evidence from open-label studies supports its long-term benefit and tolerability [45,46,47]. Potential side effects, often idiosyncratic and dose-dependent, include sedation, parkinsonism, akathisia, insomnia and particularly depression and suicidal ideation, with one complete suicide reported in the TETRA-HD study. TBZ is therefore contraindicated in individuals with poorly controlled depression or active suicidal ideation, and close psychiatric monitoring is advised [48]. Due to its short half-life (5–8 h), TBZ requires three daily doses, starting low (e.g., 12.5 mg/day) and titrating every 1–2 weeks to a maximum of 75 mg/day [49,50]; in carefully selected patients, higher doses up to 200 mg/day may be used [51]. Once clinical improvement is achieved, periodic reassessment is recommended to determine the need for ongoing therapy and gradual dose reduction should be considered [29]. The posology, including recommended starting and maximal doses of the most commonly used drugs for the treatment of chorea in HD, is presented in Table 1.

TBZ and all VMAT2 inhibitors are metabolized via the cytochrome P450 2D6 (CYP2D6). Polymorphisms and co-administration of strong CYP2D6 inhibitors (e.g., paroxetine, fluoxetine, fluvoxamine) can raise plasma levels, increasing the risk of adverse events [52]. Genotyping is not routinely necessary [48].

Deutetrabenazine (DEU) is a deuterated analogue of TBZ, in which key hydrogen atoms are replaced by deuterium (a heavy hydrogen isotype). This modification slows CYP2D6 metabolism, prolongs the half-life of active metabolites, and results in more stable plasma concentrations, allowing for twice-daily dosing and potentially improved tolerability [49,53]. DEU demonstrated efficacy and safety in the FIRST-HD trial, showing significant chorea reduction and improved quality of life with no increase in depression or suicidality compared to a placebo [54]. Long-term benefit was confirmed in the ARC-HD open-label extension [55]. The recommended maximum dose is 48 mg/day, or 36 mg/day in poor CYP2D6 metabolizers, or when using strong CYP2D6 inhibitors [56].

Valbenazine (VBZ), a valine-ester prodrug of an active DEU metabolite, offers once-daily dosing due to extended half-life. In the KINECT-HD trial, VBZ was efficacious in reducing HD chorea severity and was well tolerated, with somnolence as the most common side effect [57]. The recommended dose is 80 mg/day, and although VBZ is not yet widely available, it is considered a promising agent for patients requiring simplified regimens [49,58].

There are no head-to-head studies comparing VMAT2 inhibitors. The main differences are expected to be pharmacokinetic, with DEU and VBZ offering more practical dosing [58]. An indirect comparison of TBZ and DEU suggested that DEU may be associated with a lower incidence of depressive symptoms and somnolence, although these findings require confirmation [59]. Additionally, a small open-label study reported that switching from TBZ to DEU led to improved chorea control [55].

B.Antipsychotic Drugs (APDs)/Neuroleptics

APDs are the most prescribed drugs in HD clinical practice [60]. Although data from RCTs are limited, in expert surveys they are often selected as the first-line treatment for chorea, especially when accompanied by psychiatric symptoms such as irritability, aggression, or psychosis [38,61]. Neuroleptics vary considerably in their pharmacodynamics, particularly between typical (first-generation) and atypical (second-generation) agents. Their anti-choreic effect is mainly attributed to D2 receptor antagonism, which enhances the indirect pathway. However, D2 receptor antagonism carries the risk of extrapyramidal side effects, including parkinsonism, tardive dyskinesia and akathisia, as well as apathy and cognitive dysfunction, which are particularly concerning in HD [48].

First-generation (typical) APDs, such as haloperidol, pimozide and benzamides (e.g., tiapride, amisulpride, and sulpiride), exhibit strong D2 antagonism relative to 5-HT2A binding. It offers higher potency but also a higher risk of tardive dyskinesia, parkinsonism, sedation, and akathisia [62], which limits their use in modern practice.

Their use is based on limited evidence from small open-label studies and small trials [63,64,65,66,67,68]; moreover, methodological limitations preclude strong conclusions [69]. Tiapride remains in use in several European countries due to its relatively benign profile and is included as a first-line recommendation in recent German guidelines [70], despite its unavailability in North America. Pimozide may be reserved for refractory cases, though its evidence is largely anecdotal and its primary indication remains Tourette syndrome [68]. Haloperidol, although it has important side effects, is a practical option specially in resource-limited settings due to low cost [71].

Their use is supported by limited evidence from small open-label studies and underpowered clinical trials [63,64,65,66,67,68]; however, significant methodological limitations preclude definitive conclusions [69]. Tiapride remains in clinical use in several European countries due to its relatively favorable tolerability profile and is endorsed as a first-line treatment in recent German guidelines [70], although it is not available in North America. Pimozide may be considered in refractory cases, although supporting evidence is largely anecdotal and its primary indication remains Tourette syndrome [68]. Despite its adverse effect profile, haloperidol continues to be a practical choice in resource-limited settings due to its low cost [71].

Second-generation (atypical) antipsychotics offer a more favorable tolerability profile due to the fact that they combine moderate D2 antagonism with 5-HT2A and histaminergic modulation.

Risperidone is considered one of the most effective, likely due to its higher affinity for D2/D3 receptors. However, it is more frequently associated with extrapyramidal symptoms [72,73,74,75]. Olanzapine showed a favorable effect in two out of three small open studies [76], and is known to induce weight gain and sedation—mainly through H1 receptor antagonism. Interestingly, both effects can be beneficial in HD [77,78,79,80,81]. Aripiprazole is a partial agonist at D2 and 5HT1A, and antagonist at 5HT2A receptors [82] with preferential mesolimbic activity, and stabilizing effect on dopaminergic tone. Small clinical studies and case reports support its efficacy in chorea reduction, with additional potential benefits for apathy, sedation, and sleepiness [69,83,84,85]. An analysis from the Enroll-HD database suggests that risperidone and olanzapine may be at least as effective as TBZ in controlling chorea [74].

Other agents such as quetiapine [86,87] and clozapine [88,89] have also been used, but are less effective for chorea due to their low D2 receptor affinity.

C.Amantadine

Amantadine is a weak, noncompetitive N-methyl-D-aspartate (NMDA) receptor antagonist, originally developed as an antiviral agent, and later serendipitously found to improve dyskinesias in Parkinson’s disease. It modulates glutamatergic transmission in the basal ganglia and interacts with sigma-1, nicotinic, and 5-HT3 serotonin receptors, which may further influence dopaminergic tone and cortical excitability [90].

Clinical data are conflicting. One trial using oral amantadine up to 300 mg over two weeks found no significant changes in objective chorea scores, though some participants reported subjective benefit [91]. In contrast, a trial with doses up to 400 mg showed a 36% reduction in UHDRS chorea scores [92] without evidence of cognitive side effects, and another crossover study demonstrated benefit following both intravenous and oral administration [93]. Additionally, a small open-label study reported sustained efficacy with oral amantadine over one year of follow-up [94]. However, a Cochrane meta-analysis concluded that existing trials were underpowered and heterogeneous, and do not support consistent efficacy [76]. Nonetheless, some clinicians consider it a reasonable alternative when first-line options are not tolerated or contraindicated [38]. Common adverse effects include agitation, insomnia, peripheral edema, and livedo reticularis, particularly at higher doses.

D.Other Agents with Limited Evidence

Memantine, a low-affinity, non-competitive NMDA receptor antagonist that modulates glutamatergic excitotoxicity, showed potential benefits in a small open-label study [95], though this finding requires replication.

Riluzole is a glutamatergic modulator that inhibits presynaptic glutamate release. Initial small studies yielded inconsistent results [96,97]. However, a three-year RCT failed to demonstrate significant benefit on motor symptoms [98], and based on available evidence, it is not currently recommended for HD chorea.

Pridopidine is a small molecule with a complex mechanism of action. Initially developed as a dopamine stabilizer, it primarily acts as a highly selective sigma-1 receptor (S1R) agonist [99], a mechanism believed to underlie its proposed neuroprotective and homeostatic effects on intracellular signaling and mitochondrial function. Despite promising previous data [100,101,102], the PRIDE-HD and PROOF-HD trials did not meet primary motor endpoints [103]. Post hoc analyses, however, suggested possible motor benefit in patients not receiving concurrent dopamine-blocking agents [104]. As a strong CYP2D6 inhibitor, it may interact with VMAT2 inhibitors [105].

Other agents such as levetiracetam, valproate and clonazepam have been described in case series or case reports, though their use remains anecdotal [106,107,108].

E.Deep Brain Stimulation (DBS)

DBS targeting the pallidum has been explored as an option for patients with severe and medically refractory HD chorea. Available data from case reports, open-label studies, and small trials suggest that DBS can lead to a sustained reduction in chorea severity in carefully selected individuals, without significant deterioration in cognition or mood. However, its uncertain impact on the overall HD motor symptoms, variability in outcomes, and the progressive nature of HD remain important limitations [109,110,111,112,113,114,115,116,117,118,119]. A large international, multicenter RCT (NCT02535884) has recently been completed, and results are pending publication.

Alternative pallidal targets have been explored. One study reported greater chorea reduction with ventral GPi stimulation compared to the classical dorsal GPi target [120], while other small studies have evaluated GPe stimulation with promising results [37,119]. Further research is needed to clarify optimal targeting and patient selection criteria.

F.Non-Pharmacological Interventions

Motor rehabilitation, including physical and occupational therapy, may help reduce the functional impact of chorea through safety-focused strategies and environmental adaptations (e.g., grab bars, weighted utensils). These interventions can support daily function and mitigate the psychosocial burden of involuntary movements [121,122,123].

##### Ongoing Research

Bevantolol, a non-selective β-adrenergic antagonist initially developed as an antihypertensive, has recently gained attention as a potential anti-choreic therapy following the discovery of its VMAT2 inhibitory properties. In a recent proof-of-concept study, Bevantolol demonstrated a modest but consistent reduction in chorea severity, with a favorable tolerability profile [124]. Two phase IIb clinical trials—SOM3355 (NCT03575676 and NCT05475483)—have been completed, and results are pending publication.

##### Authors Recommendations

Pharmacological treatment should only be considered when chorea significantly interferes with function or causes social distress and therapy must be individualized based on severity and comorbidities.

TBZ remains the first-line treatment unless contraindicated by active depression or suicidality. DEU and VBZ are emerging alternatives with at least more favorable pharmacokinetic profile. Second-generation antipsychotics, especially aripiprazole or olanzapine, are appropriate when neuropsychiatric symptoms (NPSs) such as irritability or psychosis coexist, or when VMAT2 inhibitors are poorly tolerated. In patients with mild chorea or intolerance to first-line therapies, amantadine may be considered, though evidence is limited. Importantly, the clinical impact of anti-choreic therapy may become evident after 4 to 8 weeks of treatment. In severe, refractory cases, typical antipsychotics such as pimozide, tiapride or haloperidol should be considered or even pallidal DBS may be used as a palliative strategy in selected patients.

#### 3.1.2. Motor Impersistence

Motor impersistence is a core motor feature of HD, defined as the inability to maintain a voluntary muscle contraction at a constant level [17] resulting in a difficulty to sustain postures or continuous movements [125,126]. This leads to frequent compensatory repositioning or postural collapse that may affect oculomotor, facial, axial or limb musculature. It may be highly disabling, contributing to manipulation difficulties, impaired autonomous gait, and even dysphagia [17,127].

Unlike chorea, motor impersistence progresses steadily over the course of the disease, making it a possible surrogate marker of disease severity [17,128]. In addition to its functional consequences, it underlies characteristic exploratory signs, including tongue impersistence, the “milkmaid’s grip” in the hands [28,125], or the “rubbery gait” caused by loss of postural tone in the trunk and lower extremities [129].

It was firstly described in patients with right hemispheric stroke [130], and it has also been reported in PSP [131] and in moderate or severe dementia stages of Alzheimer’s Disease [132]. Although often described clinically as a form of “negative chorea”, it reflects a distinct and incompletely understood pathophysiological substrate: impaired sustained output from basal ganglia–thalamocortical circuits driven by progressive degeneration of D1- and D2-receptor-expressing MSNs, and further exacerbated by altered skeletal muscle excitability [133,134].

##### Clinical Management and Author’s Recommendations

There are no approved or specifically targeted pharmacological treatments for motor impersistence. The symptom is understudied, in part due to the lack of standardized, sensitive clinical tools to quantify its severity and monitor treatment response. A small, randomized, crossover study evaluated the effect of apomorphine in five patients and demonstrated transient improvement in motor persistence as assessed by the tongue protrusion subitem of the UHDRS-TMS. However, the absence of a clear pathophysiological rationale to explain this effect, the limited sample size and short-lasting effect do not preclude any therapeutic recommendation [135].

In clinical practice, it is often empirically managed with the same agents used for chorea, especially when both symptoms coexist. VMAT2 inhibitors, APD, or amantadine may offer indirect benefits, although specific efficacy for motor impersistence has not been systematically studied.

#### 3.1.3. Dystonia

Dystonia is reported in over 90% of HD patients, with its severity correlating positively with disease progression, and overall motor and cognitive impairment [136,137,138]. It may also be exacerbated by antidopaminergic medications.

It is defined as sustained or intermittent involuntary muscle contractions that result in twisting, repetitive movements (so called-dystonic movements, when lasting several seconds) or abnormal postures (so called-dystonic postures, when they last minutes to hours). Unlike chorea, dystonic movements are patterned and repeatedly involve the same group of muscles [28,29].

In adult-onset HD, dystonia is typically mild, generalized, and often coexists with chorea. It becomes evident during voluntary motor tasks such as tandem gait. Common manifestations include sustained fist clenching, excessive trunk or knee flexion, and abnormal arm posturing (e.g., “handbagging”). As the disease progresses, dystonia tends to become more persistent and functionally limiting [17]. Segmental and focal fixed dystonia forms are less frequent but may occur. In juvenile-onset HD, dystonia is often an early and prominent symptom.

The pathophysiology of dystonia in HD is multifactorial. Progressive degeneration of striatal MSNs, affecting both the direct and indirect pathways, leads to dysregulated output from the GPi and GPe reducing tonic inhibition of the thalamus and resulting in excessive activation of premotor areas. This is accompanied by abnormal cortical motor processing, including impaired movement preparation, hyperexcitability in premotor and supplementary motor areas, disorganized sensorimotor integration, and loss of inhibitory mechanisms. Emerging evidence also suggests the involvement of cerebellar and cortico-striato-thalamo-cortical loops, which may further impair motor program execution. Together, these alterations disrupt agonist–antagonist coordination and promote persistent dystonic motor patterns [28].

##### Clinical Management

There are no approved pharmacological treatments specifically developed for dystonia in HD. In most adult-onset cases, dystonia is mild and does not require targeted treatment. As a first step, medications that may aggravate dystonia should be reduced or discontinued. When treatment is needed, management generally follows the same principles as for isolated generalized or segmental dystonia.

Oral anticholinergics such as trihexyphenidyl have shown efficacy [139], but in HD, their use is generally discouraged due to poor tolerability and cognitive adverse effects. Baclofen, a GABA-B receptor agonist, may offer modest benefit orally, but stronger evidence supports intrathecal or intraventricular infusion in severe, refractory cases—particularly in juvenile-onset HD [140]. Long-acting benzodiazepines, such as clonazepam and diazepam, can offer symptomatic relief owing to their central muscle relaxant and anxiolytic properties, but sedation frequently limits their use. Tizanidine, an α2-adrenergic agonist, has been reported to confer modest benefits [141].

Focal dystonias, particularly those causing pain or interfering with hygiene and function in advanced stages (e.g., hand closure, cervical posturing), may respond well to botulinum toxin A injections [142,143].

Non-pharmacological interventions including active and passive physiotherapy, stretching exercises, and postural re-education are essential to maintain range of motion and prevent contractures in advanced stages or juvenile-onset HD [30,141]

DBS targeting the GPi and, more recently, the subthalamic nucleus (STN), has shown potential benefits in reducing dystonia severity and improving quality of life in small case series of juvenile-onset HD with severe, generalized, refractory dystonia. However, responses are variable, and the procedure carries non-negligible risks, particularly in the context of cognitive decline and progressive neurodegeneration [144,145].

##### Ongoing Research

Post hoc FIRST-HD trial analyses suggested that DEU may contribute to dystonia improvement, although this was not a primary endpoint [54]. Supporting this possibility, a recent open-label pilot study reported good tolerability, and potential symptomatic benefit in patients with isolated dystonia, particularly with blepharospasm [146].

##### Author’s Recommendations

In adult-onset HD, dystonia is frequent but often non-disabling during the early and mid-stages and no targeted intervention is required. In advanced or juvenile-onset HD cases, systemic agents such as benzodiazepines or baclofen may be used empirically, with caution regarding sedation and cognitive effects. In juvenile-onset HD, severe, refractory cases, intrathecal baclofen or DBS may be considered, although patient selection must account for overall disease progression and cognitive status.

#### 3.1.4. Parkinsonism

Parkinsonism is a near-universal motor feature in advanced stages of HD, manifesting with rigidity, bradykinesia, and postural instability [147]. It is worth noting that impairments in finger-tapping rhythm and speed can appear early in the disease course, reflecting deficits in fine motor control rather than true bradykinesia.

In juvenile-onset HD, as previously mentioned, parkinsonism, often accompanied by dystonia, is usually the predominant motor phenotype [25,26]. A similar phenotype, though rare, can also appear in very-late-onset HD.

The development of parkinsonism result from the progressive spread of neurodegeneration to D1-expressing MSNs, particularly those projecting to the GPi and substantia nigra pars reticulata (SNr), leading to underactivation of the direct pathway and excessive thalamic inhibition.

##### Clinical Management

When parkinsonian features emerge, it is essential to review the patient’s current medications, as commonly used drugs—including neuroleptics and VMAT2 inhibitors—can contribute to bradykinesia and rigidity.

In patients with functionally limiting symptoms, especially in juvenile-onset HD, levodopa may offer partial and transient benefits [30,148] although chorea or NPS may worsen or re-emerge. Case reports support the use of dopamine agonists [149], amantadine [150] and MAO-B inhibitors, which may also positively affect mood and apathy [151]. Importantly, MAO-B inhibitors should be avoided in patients receiving VMAT-2 due to the risk of hypertensive crisis and serotonin syndrome.

As in dystonia physiotherapy focused on joint mobility, stretching, postural training, and relaxation techniques can help mitigate stiffness, improve mobility, and prevent contractures [30].

##### Author’s Recommendations

In adult-onset HD, parkinsonism typically appears in advanced stages or is secondary to medication. In such cases, the priority is to reduce or discontinue contributing drugs, and dopaminergic therapy is rarely needed.

In contrast, parkinsonism is often a core and early manifestation of juvenile-onset HD, frequently accompanied by disabling dystonia. In these patients, levodopa or amantadine may be considered. In highly selected patients with severe treatment-refractory parkinsonism, DBS as for dystonia has been explored as a palliative strategy.

#### 3.1.5. Gait and Balance Disorders

Gait disorders in HD are complex and multifactorial, with heterogeneous clinical presentations, arising from dysfunction across striatal, cortical, cerebellar, and postural control circuits [152,153,154].

They are not solely attributable to chorea [67], but may also reflect motor impersistence, dystonia, impaired motor planning, loss of dual-tasking ability, bradykinesia, and postural instability [129].

##### Clinical Management

There are no approved pharmacological treatments specifically targeting gait dysfunction. While anti-choreic therapies may lead to modest improvements in selected patients, they may also worsen gait when used at higher doses by inducing parkinsonism or freezing of gait.

Therefore, management is primarily based on non-pharmacological strategies. Specialized physiotherapy programs [155,156], including proprioceptive training, postural transitions, and coordination exercises, combined with home-based exercise regimens [157,158] and early introduction of assistive devices—such as walkers or wheelchairs—have shown benefit in maintaining ambulation and reducing fall risk [159].

##### Ongoing Research

Although, as previously mentioned, the PRIDE-HD and PROOF-HD trials did not meet primary motor endpoints, some exploratory analyses have suggested a possible benefit of pridopidine on gait and balance subitems in early HD stages [160]

##### Author’s Recommendations

Early, sustained specialized physiotherapy, and the early use of assistive devices is the cornerstone of gait management. Control of chorea may help in some cases, but caution is advised with high-dose anti-choreic therapy, as it may paradoxically impair gait.

#### 3.1.6. Other Motor Manifestations

**Akathisia** is characterized by a subjective feeling of inner restlessness combined with involuntary, purposeless movements and an inability to remain still [29]. While it can be mistaken for chorea, it is typically iatrogenic, most often induced by TBZ, APD, or selective serotonin reuptake inhibitors (SSRIs) [51]. Management involves withdrawing the causative agent. Benzodiazepines or beta-blockers may help in refractory cases [30].

**Myoclonus** is more frequently observed in juvenile-onset HD where it may coexist with epilepsy. It typically presents as sudden, jerky, stimulus-sensitive movements, reflecting their cortical origin in HD [161]. When prominent, it may interfere with motor function and contribute to gait instability and falls. First-line treatment options include levetiracetam, clonazepam, and valproic acid [70,162].

**Motor and phonic tics**, including grunting, snorting, or guttural sounds, may be present in HD and can be difficult to differentiate from chorea [163]. Specific treatment is rarely required; when necessary, individualized management should generally follow an approach similar to that used for chorea.

**Dysphagia** is a common and clinically significant symptom in advanced stages. It increases the risk of aspiration pneumonia (the leading cause of death) and contributes to malnutrition and cachexia [164]. Management should include regular evaluation of swallowing function by a speech-language pathologist or swallowing therapist [30] and caregivers should be trained to recognize early signs of aspiration, such as coughing during meals or voice changes [30]. Management includes dietary texture modifications (e.g., pureed foods, thickened liquids) and compensatory swallowing techniques [165]. When oral intake becomes unsafe or insufficient, percutaneous endoscopic gastrostomy (PEG) feeding should be considered. Early discussion of this topic with the patient and caregivers is essential to align decisions with patient preferences [30]. Although no medications directly improve dysphagia, reducing pharyngeal chorea may indirectly facilitate safer swallowing.

**Weight loss** is multifactorial and highly prevalent in HD, resulting from hyperkinetic movements, reduced intake (due to psychiatric symptoms or dysphagia), and increased basal metabolic rate (linked to hypothalamic or mitochondrial dysfunction). A multidisciplinary approach with regular nutritional assessment and high-calorie, high-protein supplementation (if necessary) is essential [166]. H1 antagonist agents should be considered, particularly when weight loss is associated with chorea (e.g., olanzapine) or with depression or insomnia (e.g., mirtazapine).

A summary of current recommendations for the clinical management of motor symptoms other than chorea is provided in Table 2.

### 3.2. Cognitive Symptoms

Cognitive impairment is a core feature of HD, often emerging years before motor onset, and progressing steadily throughout the disease course [17]. The predominant pattern is a dysexecutive syndrome, reflecting early degeneration of fronto-striatal circuits—especially the caudate nucleus and its projections to the dorsolateral prefrontal cortex [167,168,169,170]. Clinically, this manifests as slowed information processing, reduced working memory, impaired planning, cognitive inflexibility, and deficits in social cognition [167,168,169,170]. Importantly, as previously mentioned, anosognosia is frequent and clinically significant, often impairing engagement with care and compromising safety [171,172].

As the disease advances, posterior cortical involvement becomes more prominent leading to a progressive, predominantly non-amnestic dementia [168], with early visuospatial impairment (such as impaired mental rotation) [173], and language disintegration including syntactic errors, anomia, and poor discourse cohesion [174,175]. Some cognitive alterations, such as perseverative ideation, are discussed in the neuropsychiatric section due to their overlap with behavioral symptoms.

#### 3.2.1. Clinical Management

No pharmacological agent has convincingly demonstrated efficacy in reversing, delaying or improving cognitive decline in HD. It is essential to minimize medications with potential cognitive side effects. Recent Enroll-HD data indicate that VMAT2 inhibitors and APD are associated with worsening cognitive performance, particularly processing speed, and accelerated functional decline [176].

Although degeneration of cortical cholinergic transmission has been proposed, clinical trials with cholinesterase inhibitors have yielded inconclusive or negative results. Early open-label studies with rivastigmine suggested modest benefit [177,178,179], but later trials failed to replicate these findings [180]. Galantamine [181] and donepezil [182] showed either no benefit or worsened symptoms—donepezil, in particular, was associated with increased chorea, falls, irritability, and anxiety [183]. Memantine also failed to show Efficacy [95]. A systematic review concluded that the current evidence does not support the routine use of cholinesterase inhibitors or memantine in HD [76]. Still, up to 3.2% of patients—mainly in North America—receive these agents in clinical practice [60].

Psychostimulants such as modafinil or methylphenidate have been used off-label to enhance alertness, motivation, or processing speed, but evidence remains limited [184]. and caution is required due to potential side effects.

Though, the cornerstone of cognitive management remains non-pharmacological. Cognitive stimulation strategies—including computerized executive training, group-based cognitive therapy, and external memory aids (e.g., agendas, reminder apps)—may offer temporary benefits and possibly slow decline [185,186]. A recent RCT demonstrated that both computerized cognitive training and music therapy improved global cognition, functional outcomes, and fronto-striatal connectivity [187]. In advanced stages, care should focus on structured, low-stimulus environments with clear routines, as HD patients are especially vulnerable to sensory overload and disorganization [30].

#### 3.2.2. Ongoing Research

Several compounds targeting glutamatergic or dopaminergic transmission have been evaluated with limited success. Amantadine showed modest benefits in small studies [92], while cariprazine, a dopamine D3/D2 partial agonist, was associated with cognitive improvement in a small controlled study using the Addenbrooke’s Cognitive Examination [188]. Conversely, ketamine proved to be harmful in HD [189], and dalzanemdor (SAGE-718), a novel NMDA positive allosteric modulator, failed to meet its primary endpoints in a phase 2 trial despite initial safety and tolerability signals [190]. Other agents such as latrepirdine, a multi-target molecule (mitochondrial pore blockade, NMDA/AMPA modulation, and calcium channel inhibition) initially developed for Alzheimer’s disease and pridopidine, failed to improve cognition in HD clinical trials [99,103,160,191,192,193]. Similarly, SSRIs failed to improve cognition in non-depressed HD patients [194,195].

#### 3.2.3. Author’s Recommendations

It is essential to avoid, whenever possible, medications that may worsen cognitive function, such as VMAT2 inhibitors or APD particularly in advanced stages. In cases where comorbid depression may further impair cognition, the use of vortioxetine—given its favorable cognitive profile—should be considered. In very selected cases, psychostimulants may also be explored, particularly in patients with prominent apathy or slowed processing speed.

### 3.3. Behavioral and Neuropsychiatric Symptoms (NPS)

NPS are nearly universal in HD, often emerging years before motor onset and significantly contributing to impair functional independence, cause emotional distress, and reduce quality of life for both patients and caregivers [196,197,198,199]. Their expression is heterogeneous, fluctuating, and poorly correlated with disease stage, which contributed to their exclusion from the HD-ISS biological staging system [200].

These symptoms arise from progressive neurodegeneration—particularly of frontostriatal and limbic circuits—but also reflect psychological responses to disease awareness, functional loss, or the burden of being a gene expansion carrier. Moreover, certain HD medications, particularly those targeting motor symptoms, may exacerbate psychiatric features such as apathy or depression [201].

Importantly, no pharmacological treatment has been approved for any psychiatric symptom in HD. Current management strategies are largely extrapolated from other conditions, with limited supporting evidence from case reports, or small open-label studies in HD [19,70,201]. Nonetheless, the need for targeted, symptom-specific, and individualized approaches are essential, combining pharmacological strategies with non-pharmacological interventions.

This section reviews the most clinically relevant NPS in HD (Table 3), many of which are captured by the Problem Behaviors Assessment—short version (PBA-s). Less frequent symptoms such as hallucinations or disorientation, although included in the PBA-s, are not addressed in dedicated sections due to the lack of HD-specific treatment strategies and their frequent overlap with broader cognitive or psychiatric syndromes.

#### 3.3.1. Depression

Depression affects up to 50% of individuals with HD across all the disease course [202]. The pathophysiology involves degeneration of fronto-limbic circuits, basal ganglia, and prefrontal cortex, alongside serotonergic dysregulation [203].

##### Clinical Management

As with other NPS in HD, available evidence is scarce, mostly based on case reports, open-label studies, or post hoc analyses, and lacks high-quality randomized trials [204]. SSRIs and serotonin-norepinephrine reuptake inhibitors (SNRIs) are the most commonly used pharmacological options in clinical practice [19,205] with small studies supporting the use of venlafaxine [206] and citalopram [195]. Treatment selection is often guided by the associated symptom profile: sertraline or escitalopram in cases with prominent irritability [207] or obsessive–compulsive symptoms [208]; mirtazapine (a noradrenergic and specific serotoninergic molecule) when insomnia or weight loss is present [209], and duloxetine in cases with concurrent neuropathic pain.

In patients with severe or recurrent depression, adjunctive use of mood stabilizers such as valproate, lamotrigine [210], lithium [211], or carbamazepine may be considered to reduce relapse risk [30]. Electroconvulsive therapy (ECT) has shown efficacy in treatment-resistant cases but should be reserved for exceptional situations due to the risk of short-term memory impairment [212,213,214,215].

Non-pharmacological interventions such as cognitive behavioral therapy (CBT), and supervised physical activity may improve depressive symptoms [201].

##### Author’s Recommendations

Antidepressant selection should be symptom-driven. We recommend avoiding tricyclic antidepressants, paroxetine, and fluvoxamine due to their higher anticholinergic burden and potential for cognitive side effects.

#### 3.3.2. Anxiety

Anxiety is common in HD but often underrecognized or misattributed to other behavioral changes. It frequently coexists with depression and may manifest as generalized worry, somatic tension, panic attacks, or anticipatory anxiety. It is frequently reactive to disease awareness, functional decline, and social or economic stressors [201].

Neurobiologically, it has been associated with dysregulation in fronto-limbic circuits, including the amygdala, insula, and anterior cingulate cortex, with serotonergic dysfunction likely contributing to symptom expression.

##### Clinical Management and Author’s Recommendations

Initial steps include identifying and treating psychiatric or medical comorbidities that may worsen anxiety [201]. SSRIs or SNRIs are first-line treatments, which offer anxiolytic effects while addressing concurrent depressive symptoms, with some evidence supporting fluoxetine [195]. Benzodiazepines may be considered for short-term or situational relief, particularly during SSRI/SNRI initiation, but their long-term use should be avoided due to the risk of sedation, falls, and cognitive impairment.

Non-pharmacological interventions are essential and should include structured daily routines, relaxation techniques, caregiver education, and—when cognitively feasible—CBT.

#### 3.3.3. Apathy

Apathy affects up to 90% of HD patients, is among the most disabling NPS, and the NPS most consistently associated with disease progression in large cohort studies such as TRACK-HD [196,216,217]. While insight into this symptom may be preserved in early stages, it often diminishes as the disease progresses [218].

It is defined as a state of decreased motivation resulting in reduced goal-directed activities, not explained by cognitive impairment, emotional distress, or reduced consciousness, although it often overlaps with depression and cognitive impairment [218,219,220]. Neural correlates include dysfunction in the dorsolateral, orbitofrontal and ventromedial prefrontal cortices, as well as the anterior cingulum, amygdala and ventral tegmental area [221,222,223].

##### Clinical Management

No pharmacological treatment has proven effective for apathy in the general population or HD population. Initial steps include reviewing and reducing potentially aggravating medications and addressing comorbid depression if present [201].

Non-pharmacological interventions—such as structured daily routines, caregiver coaching, and personalized social or physical engagement—are the cornerstone of management, and should be adapted to the patient’s cognitive status [201].

##### Ongoing Research

Bupropion (a norepinephrine–dopamine reuptake inhibitor) [224], aripiprazole [225], modafinil [184], and cariprazine [188] have shown signals of potential benefit in small studies, although none demonstrated clear efficacy in larger trials. SRX246 (a vasopressin V1a receptor antagonist) failed to meet endpoints in a phase 2 trial.

##### Author’s Recommendations

Based on clinical experience, if depressive symptoms are present, an SSRI should be trialed. In non-depressed but severely apathetic individuals, activating agents like modafinil or methylphenidate may be considered. Medications that may worsen apathy should be reduced or replaced by agents with a more activating profile, such as aripiprazole.

#### 3.3.4. Irritability and Aggressiveness

Irritability is defined as a proneness to anger, which can unfortunately lead to an outburst of rage and even verbal or physical aggression. Its significantly contributes to social isolation, caregiver burden and, in advanced stages, institutionalization.

It has been associated with dysfunction in fronto-limbic circuitry, including the striatum, amygdala, pulvinar and prefrontal cortex, with impaired serotonergic signaling and reduced structural integrity affecting emotional regulation and inhibitory control [226,227,228].

##### Clinical Management

SSRIs—particularly sertraline, escitalopram, and fluoxetine —are considered first-line treatments, even in non-depressed patients, as they have shown to reduce irritability and aggression, and are generally well tolerated [195,207,229,230,231]. Their efficacy may depend on reaching moderate-to-high doses, which downregulate presynaptic 5-HT1A autoreceptors and enhance serotonergic transmission in fronto-limbic circuits involved in affect regulation [231]. High-dose paroxetine and fluvoxamine should be avoided due to unfavorable adverse effects [229].

Atypical neuroleptics should be considered in severe cases that result in aggressive behavior or when chorea coexists [78,232,233]. Aripiprazole, in particular, may offer additional benefit due to its dopaminergic modulation and relatively favorable side effect profile [225,234]. In cases with emotional lability or poor behavioral control, mood stabilizers such as valproate or carbamazepine should be considered as adjunctive therapy [19,210,235].

Non-pharmacological strategies should be prioritized, even before pharmacological treatment. Behavioral interventions include avoiding confrontations, de-escalating early signs of irritability, using distraction techniques, and ensuring a structured, low-stimulus environment to reduce emotional reactivity. Measures must also ensure the safety and well-being of family members, especially when children are involved.

##### Ongoing Research

SRX246 showed good tolerability in the STAIR phase 2a trial [236], with exploratory analyses suggesting possible benefit in aggressive behavior [237]. Dextromethorphan/quinidine, used to treat pseudobulbar affect in ALS, failed to show improvement in irritability in a proof-of-concept study on 20 HD patients [238].

##### Author’s Recommendations

For mild to moderate irritability, SSRIs remain the preferred first-line treatment, often requiring doses at the higher end of the therapeutic range. In partial responders, adjunctive therapy with agents such as mirtazapine may be helpful, particularly when comorbid insomnia or weight loss is present. For severe irritability or overt aggression, atypical APDs are indicated. Treatment must always be guided by a careful risk–benefit analysis, as these symptoms may endanger the safety of the patient or others, occasionally requiring hospitalization.

#### 3.3.5. Perseverative Ideation and Obsessive–Compulsive Disorder (OCD)

Perseverative ideation and conduct are very common in HD, tend to increase as functional capacity declines, correlating with lower Total Functional Capacity (TFC) scores [196]. Perseverative cognition has been linked to increased risk of suicidal ideation, especially when combined with other NPS. This symptomatology is thought to arise from frontal–striatal circuit degeneration, particularly affecting cognitive flexibility [239].

Classic OCD is less frequent, though still more common than in the general population [196]. A key clinical distinction lies in the lack of intrusive distress: whereas OCD typically involves ego-dystonic, anxiety-provoking thoughts, perseverative patterns in HD tend to be ego-syntonic and reflect cognitive rigidity rather than true compulsivity.

##### Clinical Management and Author’s Recommendations

When pharmacological intervention is needed, SSRIs are generally considered the first-line treatment, particularly when symptoms coexist with anxiety or depression. Sertraline has shown benefit in case reports for obsessive symptoms [208]. In more severe cases, especially when symptoms are accompanied by strong irritability or poor behavioral control, adjunctive treatment with atypical antipsychotics such as olanzapine or risperidone may be useful [240].

In patients meeting criteria for classical OCD, high-dose SSRIs may be necessary. If inadequate response, clomipramine—a tricyclic antidepressant with strong serotonergic activity—may be cautiously trialed, although its anticholinergic burden and potential interactions with VMAT2 inhibitors warrant careful monitoring [241].

Non-pharmacological approaches, though also understudied, are crucial. CBT, when cognitive capacity allows, may help manage intrusive thoughts or compulsive behaviors. Psychoeducation for caregivers is essential to differentiate between behavioral rigidity and psychiatric compulsions, promoting more appropriate management strategies [30].

#### 3.3.6. Psychosis

Psychotic symptoms are relatively uncommon in HD, yet recent findings from large cohorts such as ENROLL-HD suggest a higher prevalence than classically reported [242]. Surprisingly, it has been associated with lower CAG repeat counts but younger age at diagnosis [243], and is often present in the context of prominent cognitive impairment [244].

##### Clinical Management

First-line treatment consists of atypical APD, though data supporting their use in HD are limited to small case series, observational studies and small trials [69,77,78,79,81,83,84,232,233]. To date, there is no evidence that favors one drug over another, although risperidone [75], olanzapine and aripiprazole [84] are the most commonly prescribed. Clozapine may be considered in pharmacoresistant cases or in patients who develop disabling parkinsonism or dystonia given its more favorable extrapyramidal side-effect profile.

ECT has been shown to be effective in selected cases, particularly when psychotic symptoms are severe, treatment-resistant, or coexist with mood disturbances or suicidality [212,213].

##### Author’s Recommendations

Firstly, it is essential to identify and treat comorbid medical conditions that may precipitate acute psychotic episodes, including infections, metabolic disturbances, substance use, or medication effects. In patients with established psychosis, we recommend initiating atypical antipsychotics, selecting the agent based on comorbid features, titrating slowly with close monitoring of psychiatric and motor status.

#### 3.3.7. Suicidal Ideation

Suicidality represents a major concern in HD, with suicide rates 2–6 times higher than in the general population [245,246,247]. Risk is elevated throughout the disease course but is especially high around genetic diagnosis, the initial appearance of symptoms, and when functional dependency begins [248,249]. Contributing factors include psychiatric comorbidities (e.g., depression, anxiety, impulsivity), reduced insight, social stressors, and existential distress [48]. Importantly, suicidal ideation may occur independently of major depression, often driven by impulsivity and perseverative cognition.

##### Clinical Management and Author’s Recommendations

Suicide risk should be systematically assessed in all patients with HD, regardless of disease stage. Prevention involves prompt identification and treating risk factors such as underlying depression, impulsivity, and social isolation [30]. As previously noted, some drugs such as TBZ may worsen mood or induce suicidality in vulnerable individuals [250], warranting close monitoring during dose titration.

Although evidence is limited, one open-label study suggested a reduction in suicidal ideation following treatment with olanzapine [240]. In high-risk scenarios, caregivers’ involvement in safety planning is essential, and hospitalization should be considered when safety cannot be assured.

#### 3.3.8. Impulsivity

Impulsivity refers to the tendency to act prematurely without adequate forethought, often leading to socially inappropriate comments, emotional outbursts, compulsive shopping, disinhibited sexual behavior, or other high-risk actions. When combined with irritability, perseveration or depression, impulsivity may increase the risk of harmful behaviors, including suicide attempts. This symptom is thought to reflect orbitofrontal cortex dysfunction and its connections with the ventral striatum and limbic structures, leading to a loss of cortical inhibitory control.

##### Clinical Management and Author’s Recommendations

Impulsivity is often addressed indirectly by treating comorbid conditions. SSRIs are used when impulsivity occurs in the context of depression or irritability, while atypical APDs are indicated in cases of impulsive aggressivity. Mood stabilizers such as valproate or carbamazepine may also be helpful in flattening affective instability and reducing emotionally driven impulsive episodes [30].

Psychoeducation is essential to help caregivers understand that impulsive behavior is a neuropsychiatric symptom—not simply inappropriate conduct—and to provide them with strategies to reduce environmental triggers.

#### 3.3.9. Other NPSs

**Sleep disorders** affect two-thirds of HD patients and may include insomnia, circadian rhythm disruption, fragmented sleep, or hypersomnia [251,252]. They result from a combination of neurodegenerative changes, psychiatric comorbidities (e.g., depression, anxiety), or side effects of medications. Management should begin with identifying and addressing the underlying cause (e.g., mood disorder, nighttime movements, pain), followed by implementation of sleep hygiene measures. When necessary, short-acting hypnotics or sedative antidepressants such as mirtazapine or trazodone may be used, always with caution in patients at risk of falls or cognitive impairment. Melatonin is a pharmacologic option particularly when there is pattern of circadian rhythm disordered sleep [201].

**Sexual disorders** range from reduced libido (often associated with depression or SSRIs) to disinhibited or hypersexual behavior, which may be part of a broader impulse control disturbance. As in other symptoms, treatment should be individualized and consider psychiatric comorbidities, family distress, and the patient’s safety and dignity.

## 4. Conclusions

Symptomatic management remains the cornerstone of care in HD, despite recent advances in disease-modifying therapies. A deeper understanding of HD’s phenotypic heterogeneity and underlying pathophysiology is essential to develop effective, patient-centered treatments. Chorea is the only symptom with specifically approved pharmacological options—VMAT2 inhibitors—though their use is limited by psychiatric adverse effects. Neuroleptics are commonly prescribed, especially when psychiatric symptoms coexist, but may exacerbate cognitive and motor dysfunction. In contrast, other disabling motor features, such as motor impersistence, remain underexplored and lack targeted therapies. No pharmacological intervention has demonstrated clear efficacy in treating cognitive impairment, and psychiatric symptoms are managed based on extrapolation from other disorders, primarily using SSRIs and atypical antipsychotics. Special attention must be given to the emergence of suicidal ideation, and psychological support should be provided to both patients and their families. Overall, given the modest efficacy and frequent side effects of available medications, non-pharmacological and multidisciplinary approaches are essential to preserve function and enhance quality of life. Symptomatic care in HD must remain proactive, person-centered, and responsive to the complex and evolving needs of each individual and their families.

## Figures and Tables

**Figure 1 ijms-26-06220-f001:**
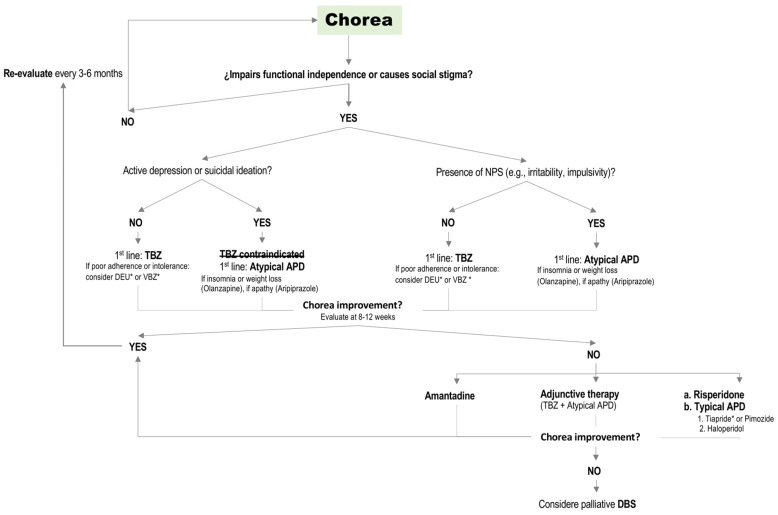
Flowchart illustrating the decision tree for recommended treatments in Huntington’s disease (HD). * If available. Abbreviations: NPS, neuropsychiatric symptoms; TBZ, tetrabenazine; DEU, Deutetrabenazine; VBZ, Valbenazine; VMAT2, vesicular monoamine transporter type 2 inhibitors; APD, antipsychotic drugs; DBS, deep brain stimulation.

**Table 1 ijms-26-06220-t001:** Summary of pharmacological treatments commonly used for chorea in Huntington’s disease, including recommended starting dose, maximal dose, availability, and estimated cost. * Maximum dose of VMAT2 inhibitors may vary depending on CYP2D6 metabolizer status and tolerability. † In very selected clinical cases, TBZ doses until 200 mg/day could be considered with close monitoring. Cost classification: low—generic medication widely available; estimated cost < EUR 20/month; low–moderate—estimated cost EUR 20–100/month; high—specialty drugs with limited access or requiring prior authorization; estimated cost >EUR 500/month. Abbreviations: VMAT2 inhibitors, vesicular monoamine transporter 2 inhibitors; APDs, antipsychotic drugs; NMDA, N-methyl-D-aspartate; QD, once daily; BID, twice daily; TID, three times daily.

Mechanism	Drug Name	Recommended Starting Dose	Recommended Maximal Dose	Availability	Estimated Price
**VMAT 2 INHIBITORS**	**Tetrabenazine (TBZ)**	12.5–25 mg (QD or BID)	*^,†^ 75 mg/d (TID)	Worldwide	Low–moderate
	**Deutetrabenazine (DEU)**	12 mg (BID)	* 48 mg/d (BID)	USA, Canada, Australia, selected European countries	High
	**Valbenazine (VBZ)**	40 mg (QD)	* 80 mg/d (QD)	USA	High
**ATYPICAL APDs**	**Olanzapine**	2.5–5 mg (QD or BID)	20 mg/d (BID)	Worldwide	Low–moderate
	**Aripiprazole**	2–15 mg (QD or BID)	30 mg/d (BID)	Worldwide	Low–moderate
	**Risperidone**	0.5–2 mg (QD or BID)	16 mg/d (BID)	Worldwide	Low
**TYPICAL APDs**	**Tiapride**	50–200 mg (QD or BID)	900 mg/d (TID)	Europe	Low
	**Pimozide**	1 mg (QD)	2–4 mg/d (QD)	Worldwide (limited availability)	Low
	**Haloperidol**	0.5–2 mg (QD or BID)	10 mg/d (BID)	Worldwide	Low
**NMDA** **ANTAGONIST** **RECEPTOR**	**Amantadine**	100–200 mg (QD or BID)	400 mg (BID)	Worldwide	Low

**Table 2 ijms-26-06220-t002:** Treatment recommendations for motor symptoms (other than chorea) in Huntington’s disease: insights from published literature, expert consensus, and clinical experience. Abbreviations: TBZ, tetrabenazine; APDs, antipsychotic drugs; JHD, juvenile-onset Huntington’s disease; GPi/STN-DBS, internal globus pallidus or subthalamic nucleus deep brain stimulation; SSRIs, selective serotonin reuptake inhibitors; BZDs, benzodiazepines; LEV, levetiracetam; CZP, clonazepam; VPA, valproic acid; TS, Tourette syndrome; CBT, cognitive behavioral therapy; PEG, percutaneous endoscopic gastrostomy; H1 antagonist agents, histamine H1 receptor antagonist agents.

Motor Symptom	Special Considerations	Pharmacological Interventions	Non-Pharmacological Interventions
**MOTOR** **IMPERSISTENCE**	May be highly disabling.	No approved therapies. Empirical use of TBZ, amantadine, or APD (especially when coexists with chorea).	No specific evidence available; physiotherapy and task-specific motor training may help.
**DYSTONIA**	Often mild and action-induced; specific pharmacological treatment is rarely needed. Severe or fixed forms are frequent in JHD.	First step: reduce antidopaminergic medication if present. Second step: Same management as for idiopathic dystonias. JHD: Intrathecal baclofen or GPi/STN-DBS in refractory cases.	Active/passive physiotherapy, stretching, and postural re-education.
**PARKINSONISM**	Typically emerges in late-stage or as drug-induced; specific pharmacological treatment is rarely needed. Prominent and disabling in JHD.	First step: reduce antidopaminergic medication if present. Step step: consider levodopa or amantadine (especially if chorea coexists).	Same as for dystonia.
**GAIT** **DISTURBANCES**	Multifactorial origin; not solely attributable to chorea.	No approved therapies. Treat contributing factors (e.g., chorea, dystonia, motor impersistence) if present and functionally impairing.	Specialized physiotherapy, postural training, proprioceptive strategies, and early use of assistive devices (e.g., walker, wheelchair).
**AKATHISIA**	Mostly iatrogenic (TBZ, APD or SSRIs)	First step: identify and reduce or withdraw causative agent. Alternatives: Beta-blockers, BZD.	No specific evidence available; management is primarily pharmacological.
**MYOCLONUS**	Treatment only required if functionally disabling. Common in JHD.	First step: LEV, CZP Alternatives: VPA	No specific evidence available; management is primarily pharmacological.
**TICS**	Specific treatment is rarely needed.	Usually managed similarly to chorea.	If intervention is required, follow TS management principles: behavioral therapy (CBT/habit reversal), education, reassurance.
**DYSPHAGIA**	Clinically significant in advanced stages. Early discussions about PEG placement are essential to align care with patient preferences.	Consider treating contributing factors (e.g., pharyngeal chorea or tics).	Regular swallowing evaluations by specialists; caregiver training to detect early signs; dietary texture modifications; PEG when indicated.
**WEIGHT LOSS**	Highly prevalent and multifactorial.	High-calorie, high-protein nutritional supplementation. Consider H1 antagonist agents particularly when weight loss is associated with chorea (e.g., olanzapine), or with depression or insomnia (e.g., mirtazapine).	Regular nutritional assessment.

**Table 3 ijms-26-06220-t003:** Treatment recommendations for behavioral and neuropsychiatric symptoms in Huntington’s disease (HD), based on published literature, expert consensus, and clinical experience. When applicable, the first-line treatment recommended by the authors is indicated in parentheses. Abbreviations: NPS, neuropsychiatric symptoms; SSRIs, selective serotonin reuptake inhibitors; SNRIs, serotonin–norepinephrine reuptake inhibitors; VPA, valproic acid; CBZ, carbamazepine; LTG, lamotrigine; ECT, electroconvulsive therapy; BZDs, benzodiazepines; APDs, antipsychotic drugs; CBT, cognitive behavioral therapy; TBZ, tetrabenazine.

NPS	Special Considerations	Pharmacological Interventions	Non-Pharmacological Interventions
**DEPRESSION**	Treatment should be tailored to coexisting symptoms (e.g., irritability, apathy, insomnia).	First step: SSRIs (sertraline, citalopram) or SNRIs (venlafaxine, duloxetine). In severe or recurrent cases: consider mood stabilizers (VPA, CBZ, LTG).	Psychotherapy and/or CBT. Consider ECT in selected treatment-resistant cases.
**ANXIETY**	May be exacerbated by comorbidities or medications.	First step: address comorbid conditions. Second step: SSRIs or SNRIs Consider BZD for acute anxiety episodes or during SSRI/SNRI titration.	Structured routines, relaxation techniques, psychoeducation (for patients and caregivers).
**APATHY**	One of the most disabling and underrecognized NPS.	**No approved therapies**. First step: review and reduce potentially aggravating medications. Second step: Consider SSRIs (if depression is present), or activating agents (methylphenidate, modafinil).	Structured daily routines, and tailored social or physical activities to enhance motivation.
**IRRITABILITY AND** **AGGRESSIVENESS**	Treatment should be guided by symptom severity and coexisting psychiatric features.	First step: High-dose SSRIs (sertraline and escitalopram). Second step: APD (aripiprazole or olanzapine). Consider adjunctive mood stabilizers in refractory cases.	Behavioral strategies: avoid confrontation, de-escalate early signs, and use distraction techniques.
**PERSEVERATIVE** **IDEATION AND OCD**	Perseveration is much more common.	Perseveration: SSRIs or APD (olanzapine or risperidone). OCD: High-dose SSRIs or clomipramine.	CBT and psychoeducation (for patients and caregivers).
**PSYCHOSIS**	Important to rule out and treat comorbid medical conditions that may trigger or worsen psychosis.	First step: APD (olanzapine, risperidone or aripiprazole). If no response, consider other APDs such as clozapine.	Consider ECT in severe treatment-resistant cases.
**SUICIDAL** **IDEATION**	**Suicide risk should be systematically assessed.** May occur independently of major depression.	First step: Prompt identification and management of contributing factors (depression, impulsivity or perseveration) or withdraw possible contributing medications (e.g., TBZ) Second step: Consider APD (olanzapine)	Ensure caregiver involvement, prevent social isolation. Consider psychiatric hospitalization if safety is compromised.
**IMPULSIVITY**	Treatment should be tailored to symptom profile.	If comorbid depression or irritability: SSRIs. If impulsive aggressivity: atypical APD. Consider adjunctive mood stabilizers.	Psychoeducation (for patients and caregivers) to provide insight and reduce triggers.

## Data Availability

All data are included in the publication.

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
