# Peer review of "Update on the Symptomatic Treatment of Huntington’s Disease: From Pathophysiology to Clinical Practice"

_ijms, 2025, doi:10.3390/ijms26136220_

Round 1
Reviewer 1 Report
Comments and Suggestions for Authors
The manuscript of Olmedo-Saura et al. is a comprehensive review of the current symptomatic treatment strategies of neurodegenerative autosomal dominant Huntington disease (HD). The review describes the pathophysiological basis of each symptom and the clinical study findings for each medicinal substance. Furthemore, the review critically discusses the best available individualized management for each patient. The manuscript is a well-written and appropriately referenced broad and practical narrative review for HD that merits publication if some minor revisions are made.
Points of minor revision
- In Introduction, the current internationally accepted criteria for diagnosis of HD should be mentioned.
- A short discussion regarding the medical and ethical issues regarding treatment of presymptomatic cases after a positive genetic test should be included.
Author Response
Comment 1: In Introduction, the current internationally accepted criteria for diagnosis of HD should be mentioned.
Response 1: We appreciate this important suggestion. In response, we have added a concise summary of the internationally accepted diagnostic criteria for Huntington’s disease to the Introduction. Specifically, we now describe the classic clinical diagnostic approach—based on the presence of unequivocal motor signs. We also introduce the more recent Huntington’s Disease Integrated Staging System (HD-ISS), which enables biological staging based on clinical and imaging biomarkers. This addition improves the contextual understanding of diagnostic and staging frameworks relevant to symptomatic treatment decisions.
(See revised text in the Introduction, paragraph 3.)
Comment 2: A short discussion regarding the medical and ethical issues regarding treatment of presymptomatic cases after a positive genetic test should be included.
Response 2: We thank the reviewer for this thoughtful comment. Ethical considerations regarding early intervention in genetically confirmed but asymptomatic individuals are indeed highly relevant, particularly in the context of emerging disease-modifying strategies and the challenges associated with enrolling this population in clinical trials. As this review focuses specifically on symptomatic treatment, we initially considered this topic to be outside the primary scope of the manuscript. Nevertheless, we do address the management and follow-up of recently diagnosed, asymptomatic gene expansion carriers—particularly within the section on neuropsychiatric symptoms—where we emphasize the importance of providing early psychological support and counseling following predictive genetic testing. Even in the absence of overt symptoms, attention to mental health and early engagement with care are essential components of a comprehensive approach in this population.
Reviewer 2 Report
Comments and Suggestions for Authors
- line 464: The sentence is short. Should be complemented by some information about pridopidine’s action as a sigma-1 receptor agonist, which may support neuroprotective pathways and enhance motor control.

Only or two corretions are necessary (namely the point #1)
Author Response
Comment 1: Ln. 200: “their use stems from small open-label studies and small trials”. Seems a little awkward in terms of grammatical construction. You may consider to replace it by (e.g.):”Their use is based on limited evidence from small open-label studies and preliminary trials…”
Response 1: Thank you for highlighting this issue. We have rephrased the sentence for improved clarity. See Ln. 211: "Their use is based on limited evidence from small open-label studies and small trials; moreover, methodological limitations preclude strong conclusions."
Comment 2: Ln. 464: The sentence is short. Should be complemented by some information about pridopidine’s action as a sigma-1 receptor agonist, which may support neuroprotective pathways and enhance motor control.
Response 2: We thank the reviewer for this observation. A detailed description of pridopidine’s mechanism of action, including its role as a sigma-1 receptor agonist and its proposed neuroprotective effects, is already provided when the drug is first introduced in the manuscript (ln. 280). This section has now been further expanded and clarified. For that reason, we did not repeat the mechanistic explanation in the subsequent two mentions of pridopidine later in the text, in order to avoid unnecessary repetition.
Comment 3: Ln. 583: The abbreviature ISS wasn’t written in full text.
Response 3: We appreciate the reviewer pointing this out. We have corrected this by writing out the full term—Huntington’s Disease Integrated Staging System (HD-ISS)—at its first appearance in the manuscript. See ln. 42